# Individualization of Mycophenolic Acid Therapy through Pharmacogenetic, Pharmacokinetic and Pharmacodynamic Testing

**DOI:** 10.3390/biomedicines10112882

**Published:** 2022-11-10

**Authors:** Wolfgang Winnicki, Andreas Fichtenbaum, Goran Mitulovič, Harald Herkner, Florina Regele, Michael Baier, Sieglinde Zelzer, Ludwig Wagner, Guerkan Sengoelge

**Affiliations:** 1Department of Medicine III, Division of Nephrology and Dialysis, Medical University of Vienna, 1090 Vienna, Austria; 2Department of Laboratory Medicine, Medical University of Vienna, 1090 Vienna, Austria; 3Department of Emergency Medicine, Medical University of Vienna, 1090 Vienna, Austria; 4Clinical Institute of Medical and Chemical Laboratory Diagnostics, Medical University of Graz, 8036 Graz, Austria

**Keywords:** inosine monophosphate dehydrogenase, mycophenolic acid, kidney transplantation, allograft rejection, pharmacogenetics, pharmacokinetics, pharmacodynamics

## Abstract

Mycophenolic acid (MPA) is a widely used immunosuppressive agent and exerts its effect by inhibiting inosine 5′-monophosphate dehydrogenase (IMPDH), the main regulating enzyme of purine metabolism. However, significant unexplained differences in the efficacy and tolerability of MPA therapy pose a clinical challenge. Therefore, broad pharmacogenetic, pharmacokinetic, and pharmacodynamic approaches are needed to individualize MPA therapy. In this prospective cohort study including 277 renal transplant recipients, IMPDH2 rs11706052 SNP status was assessed by genetic sequencing, and plasma MPA trough levels were determined by HPLC and IMPDH enzyme activity in peripheral blood mononuclear cells (PBMCs) by liquid chromatography–mass spectrometry. Among the 277 patients, 84 were identified with episodes of biopsy-proven rejection (BPR). No association was found between rs11706052 SNP status and graft rejection (OR 1.808, and 95% CI, 0.939 to 3.479; *p* = 0.076). Furthermore, there was no association between MPA plasma levels and BPR (*p* = 0.69). However, the patients with graft rejection had a significantly higher predose IMPDH activity in PBMCs compared to the controls without rejection at the time of biopsy (110.1 ± 50.2 vs. 95.2 ± 45.4 pmol/h; *p* = 0.001), and relative to the baseline IMPDH activity before transplantation (*p* = 0.042). Our results suggest that individualization of MPA therapy, particularly through pharmacodynamic monitoring of IMPDH activity in PBMCs, has the potential to improve the clinical outcomes of transplant patients.

## 1. Introduction

Mycophenolic acid (MPA) as an antiproliferative agent has become an essential component of immunosuppressive therapy in a wide spectrum of indications, ranging from transplant medicine to various autoimmune diseases [1,2,3,4]. It is a selective, non-competitive, and reversible inhibitor of the enzyme inosine 5′-monophosphate dehydrogenase (IMPDH), which is crucial for the metabolization of inosine 5′-monophosphate (IMP) into xanthosine 5′-monophosphate (XMP) at the key site of the de novo synthesis pathway of guanine nucleotides [5]. Inhibiting the formation of guanosine monophosphate (GMP) results in the depletion of guanosine triphosphate (GTP) and deoxy-GTP (dGTP), which represent the substrates for DNA, RNA, protein, and glycoprotein syntheses [6]. The cytostatic activity of mycophenolic acid is more potent in lymphocytes, due to its strict dependence on the de novo purine pathway, than in other cell types, which are able to utilize a salvage pathway to generate guanosine nucleotides [7,8,9,10].

Another factor leading to MPA’s strong effect on lymphocytes is the distribution and characteristics of the two IMPDH isoenzymes, IMPDH1 and IMPDH2. For immunosuppressive effects, IMPDH2 is of particular importance. This isoenzyme is the predominant isoform in dividing cells, such as lymphocytes, [11], and it is suppressed by MPA at 4.8 times stronger than IMPDH1 [8].

The IMPDH2 enzyme is encoded by a small gene of 5.8 kilobase [12] consisting of 14 exons [13]. In a previous study, we showed the functional relevance of the intronic IMPDH2 single-nucleotide polymorphism rs11706052, which lowered the antiproliferative activity of MPA in lymphocytes by half [5].

With mycophenolate mofetil and enteric-coated mycophenolate sodium, there are two mycophenolic acid precursors available. Mycophenolic acid in general has become the immunosuppressant of choice in transplant regimens. Over seventy percent of renal and cardiac transplant recipients and over 50% of lung transplant patients receive MPA as one component of their mostly triple immunosuppressive therapy [14]. 

Although MPA is well-established in immunosuppressive treatment, a proportion of subjects do not profit from its use. Whereas some patients are complete non-responders, others require higher doses of MPA to respond appropriately. Likewise, some patients have more side effects and dose reductions are needed more frequently [15,16,17,18,19,20,21,22,23,24]. Mycophenolic acid therapy is currently administered to all adult transplant patients at standard dosages according to the manufacturer’s guidelines. However, under- or over-immunosuppression may be caused in some patients when MPA is administered in a fixed dose. Therefore, the monitoring of MPA exposure and IMPDH activity with subsequent MPA dose adjustment seems to have a promising potential to improve clinical outcomes [14].

There is a significant variability in MPA pharmacokinetics between kidney transplant recipients, with up to 10-fold difference in the MPA AUC at the same dosage of MPA. This explains the considerable interindividual differences in drug tolerability and efficacy [15,16].

In addition, there have been numerous studies on MPA efficacy that have addressed various aspects, such as genetic variability [1,5,25,26,27], gender [28], ethnicity [29], as well as drug interactions and accompanying conditions [22,30,31,32,33,34,35]. However, there are no conclusive data to date, including pharmacogenetic analyses, therapeutic drug monitoring of MPA plasma levels, and pharmacodynamic monitoring of IMPDH enzyme activity, to fully elucidate treatment response in patients receiving MPA therapy [15,36,37,38,39,40].

In this study, we hypothesized that a multimodal view of IMPDH2 SNP analysis and MPA plasma trough levels, as well as PBMC IMPDH activity measurement, allows the identification of patients who do not respond adequately to MPA therapy: the presence of the IMPDH2 SNP rs11706052, low MPA plasma trough levels, and high PBMC IMPDH activity might explain a higher rate of biopsy-proven rejections (BPR).

The study is intended to provide further evidence on whether individualization of MPA therapy is possible.

## 2. Methods and Patients

### 2.1. Study Design

This is an in vivo prospective cohort study with renal transplant patients. The primary endpoint was defined as the presence of a biopsy-proven rejection (BPR), and associations between BPR and pharmacogenetic, pharmacokinetic, and pharmacodynamic factors were analyzed. All data relevant to the study were obtained from the Medical University of Vienna, which maintains a Division of Nephrology and Dialysis and a Division of Transplantation.

### 2.2. Study Population

In this study, 277 renal transplant recipients were included. The exclusion criteria for the study participants were age under 18 years, pregnancy, and known allergies or contraindication to the immunosuppressive agents in use. Blood was drawn in the first year after transplantation at six time points (1st day before MPA started and 1, 3, 6, 9, and 12 months thereafter) during routine control at the Division of Nephrology and Dialysis of the Medical University of Vienna and, additionally, at the time of graft biopsies. Blood sampling took place predose at 12 h after the last MPA intake. MPA was taken orally at the standard dose as enteric-coated mycophenolate sodium at 720 mg (Myfortic^®^ 360 mg tablets, Novartis Pharma AG, Basel, Switzerland) or as an equivalent dose of mycophenolate mofetil at 1000 mg (CellCept^®^ 500 mg tablets, Hoffmann La Roche, Basel, Switzerland) every 12 h. The patients started immunosuppressive treatment right after the first blood draw (to determine baseline/pretransplant IMPDH activity) immediately before transplantation. Dose reductions were applied in the course when clinically indicated. Blood was used for genetic analysis, for the measurement of mycophenolic acid trough levels in plasma, and for the determination of IMPDH enzyme activity in PBMC lysates.

The demographics of the 277 kidney allograft recipients, with categorization of patients with or without BPR, are shown in Table 1.

### 2.3. Laboratory Diagnostics for Routine Parameters

All routine laboratory parameters were analyzed at the Department of Laboratory Medicine, Medical University of Vienna, which has a certified (ISO 9001:2015) and accredited (ISO 15189:2012) quality management system.

### 2.4. DNA Sequencing

To identify the IMPDH2 rs11706052 SNP among the 277 kidney transplant patients, a PCR-based dideoxy chain termination sequencing was performed using specific primer pairs with M13 tails. The DNA sequencing method used is described in detail in a previous study [5].

### 2.5. Isolating Peripheral Blood Mononuclear Cells

Peripheral mononuclear blood cells were isolated from EDTA-anticoagulated blood using Ficoll-Paque ^TM^ Plus (GE Healthcare, Uppsala, Sweden). The method of density gradient centrifugation was performed according to the protocol of our previous study [5]. Lymphocytes accounted for more than 85% of the PBMCs in repeated blood smears. Cell suspensions were dissolved in Aqua bidest to obtain a final concentration of 1 × 10^5^ cells/µL^−1^. The PBMC samples were frozen at −80 °C until further processing.

### 2.6. Determination of MPA Plasma Levels

Mycophenolic acid (MPA) concentrations in plasma were determined using a modified reversed-phase, high-performance liquid chromatographic method (HPLC) published in 2004 by Khoschsorur et al. [41]. Briefly, after protein precipitation with acetonitrile containing phosphoric acid (85%) and the internal standard Mephenytoin, MPA was separated on a Chromolith Performance RP18e column (100 × 4.6 mm, Agilent, Palo Alto, CA, USA), with a mobile phase containing 40 mM of phosphate buffer, pH 3.0, and acetonitrile (32:68 (*v*/*v*)), and measured with UV-detection at 215 nm. The intra-assay and inter-assay variations of the MPA concentrations (0.75, 1, 5, 10, and 18.5 g/mL) were 5.33–5.96% and 1.83–5.88%, respectively.

### 2.7. Determination of IMPDH Activity in PBMC

#### 2.7.1. Reagents

The calibrators, the quality controls (QC), and the reaction mixture (RM), consisting of 7.5 µL of the incubation buffer, 1.25 µL of 6 nmol/L inosine 5’-monophosphate (IMP; 6 nM), 1.25 µL of 6 nmol/L nicotine amide dinucleotide (NAD; 6 nM), and 5 µL of HPLC grade water at a total volume of 15 µL, were prepared, aliquoted, and stored at −80 °C until further use. A 7.5 mmol/L solution of XMP was diluted at a ratio of 1:250 in order to obtain the highest XMP calibrator concentration (S6). A serial dilution of S6 was performed to generate six reference concentrations (S1–S6) of XMP, consisting of 11.7 pmol (S1), 23.4 pmol (S2), 46.9 pmol (S3), 93.8 pmol (S4), 187.5 pmol (S5), and 375 pmol (S6). The incubation mixture for the IMPDH assay consisted of 12.5 µL of the sample or S1–S6 for the calibration curve, and 15 µL of the RM, 6.25 µL of HPLC grade water, and 3.125 µL of the 2.5 mol/L perchloric acid. A pool of random sample supernatants served as the inter-assay quality controls. The QC mean target value of 78.3 pmol (CV < 5%) was evaluated throughout 70 assays in a time range of 8 months. 

#### 2.7.2. Sample Preparation

The samples and working solutions were thawed and all of the following steps were performed on ice. The samples containing cell lysate were gently mixed and centrifuged for five minutes at 4 °C, 24,000× *g* rcf (Hettich Mikro 22R, Tuttlingen, Germany). The incubation mixture (IM) was assessed by transferring 12.5 µL of the supernatant into PCR strips, according to the pipetting scheme in the Appendix A. The scheme indicates identical sample treatment for the samples without incubation (t0) and with 180 min of incubation (t180). Incubation (t180) started with the addition of 15 µL of the RM into the 48 wells of rows 7–12. The PCR strips were sealed and put into the cycler block for three hours at a constant temperature of 37 °C. In the meanwhile, 24.4 µL of pre-mixed reaction mixture, water, and perchloric acid were added to the samples, which was defined as “t_0_”. After three hours of incubation, the IMPDH-mediated XMP generation was stopped by the addition of 9.4 µL of pre-mixed water and perchloric acid.

The PCR strips were centrifuged (Hettich Universal 32 R, Tuttlingen, Germany) at 4 °C at a maximum rotational speed for 5 min. The supernatant was transferred into a 96-well plate, diluted at 1:10, and mixed (using an 8-channel pipette) with mobile phase A (MoPA: 20 mol/L of ammonium formate at pH 3.45). The plate was sealed and submitted to LC–MS analysis. 

#### 2.7.3. LC–MS Analysis

HPLC gradient separation was performed using an U3000 RSLC nano HPLC system (Thermo Scientific, San Jose, CA, USA) employing a Luna Omega Polar C18, (100 Å, 1.6 µm particle size, 100 × 1.0 mm) separation column (Phenomenex, Torrance, CA, USA). The column temperature was adjusted to 18 °C in a thermostat-controlled cool water bath, and the flow rate was set to 180 µL/min. Upon the injection of 50 µL of the sample, using a 50 µL sample loop and a User-Defined Injection Program, the switching valve was set to waste position for three minutes in order to prevent contamination of the mass spectrometer with non-volatile buffer salts from the sample incubation buffer. The column equilibration conditions were as follows: 97% MoPA and 3% mobile phase C (MoPC: 80% of acetonitrile, 20% of 100 mol/L of ammonium formate, pH 3.45), and two gradients were performed within 9 min of total run time. In the first gradient, the amount of mobile phase B (MoPB: 100 mol/L of ammonium formate at pH 3.45) increased from 0–97% (3% MoPC) within one minute, reached a plateau for one minute, and decreased again within one minute to the equilibration conditions. The second gradient started at the third minute and the amount of MoPC increased to 100% within 0.5 min, reached a plateau for one minute, and decreased again to the equilibration conditions within 0.5 min.

A mass spectrometric acquisition using a maXis, Time-of-Flight mass spectrometer (Bruker, Bremen, Germany) equipped with an Apollo electrospray ion source started three minutes after the sample injection. Three technical replicates were performed for each sample, and only MS1 (no fragmentation) spectra were acquired for the target mass of the generated XMP [M+H]+ ion (*m*/*z* 365 Da). The MS was operated in a positive ion mode, and the scan range was set to 200–700 *m*/*z*. 

#### 2.7.4. Data Analysis

The target peak of the XMP [M+H]+ ion (m/z 365 Da) was integrated using “Bruker Compass Data analysis 4.2” and exported into “.csv” file format containing the following parameters: sample name, peak area (XMP), number of replicate (1–3), and time point of incubation (t_0_, t_180_). 

A XMP calibration curve was created using “MS Excel 2013”. The sample data and calibration parameters were imported into a Filemaker database (Filemaker Pro 14, Santa Clara, CA, USA). The XMP concentration at time point t0 was subtracted from the value of the t180-sample and divided by 3 to obtain “pmol/h” values for each technical replicate. The final result was the mean value of all three replicates.

### 2.8. Statistical Analysis

Continuous data were analyzed as mean ± standard deviation, and categorical data were analyzed as absolute count and relative frequency. To compare the baseline variables between the study groups, the Mann–Whitney U test, median test, or Fisher’s exact test was used.

We assessed the association between SNP (wild-type, heterozygous, and homozygous) and the presence of graft rejection using random effect logistic regression and reported the regression coefficients with 95% confidence intervals. The likelihood-ratio test was used to test for deviation from linearity. We used the same modelling approach to investigate the association between MPA or IMPDH activity and the presence of graft rejection. Using stratification and the likelihood-ratio test, we investigated an interaction between SNP and the associations between MPA/IMPDH levels. We adjusted these models for clinically chosen potential confounders, including sex, age, presence of donor-specific antibodies, and number of pre-existing renal transplantations. The standard metrics for diagnostic test accuracy were calculated, comprising the specificity and sensitivity of the predose IMPDH levels for an independent detection of the subjects with BPR. We assessed the optimal cut-off level by maximizing the sensitivity/specificity product according to Liu´s method [42] and using the adjustment approach suggested by Fluss et al. [43].

For data analysis, we used MS Excel and Stata 14 for Mac (Stata Corp., College Station, TX, USA). Generally, a two-sided *p*-value < 0.05 was considered statistically significant.

## 3. Results

In this prospective cohort study, pharmacogenetic, pharmacokinetic, and pharmacodynamic parameters were examined in 277 renal transplant patients to analyze the associations between biopsy-proven kidney graft rejection and the presence of rs11706052 SNP status, MPA trough levels in plasma, and IMPDH activity in PBMCs. The baseline characteristics of the study patients are shown in Table 1. The results of the pharmacogenetic, pharmacokinetic, and pharmacodynamic testing are presented in Table 2.

### 3.1. Graft Rejections in the Study Population

Out of the 277 renal transplant patients, 84 patients experienced graft rejections. In total, 92 episodes of graft rejection (51 borderline lesions, 33 T cell-mediated rejections, and 8 antibody-mediated rejections) were identified in 84 patients. Pathological biopsy results, including renal graft rejections according to the Banff criteria, of the study patients are presented in Appendix A.

### 3.2. Pharmacogenetic Testing

#### IMPDH2 rs11706052 SNP Status and Graft Rejection

The IMPDH2 rs11706052 SNP status in the patients with and without renal graft rejection is shown in Table 2.

With the occurrence of the SNP rs11706052 (heterozygous and homozygous), the odds ratio for graft rejection increased by 1.808 (95% CI 0.939 to 3.479; *p* = 0.076). This was not statistically significant. However, there was a significant result when a trend test was performed (*p* = 0.035). The significance of the trend test did not change irrespective of the exclusion of borderline rejections (*p* = 0.034). The results indicated a clear trend showing the association of the rs11706052 SNP status with kidney graft rejection risk, but the sample size in the overall analysis was too small to show statistical significance. 

### 3.3. Pharmacokinetic Testing

#### 3.3.1. Oral MPA Dosage and MPA Plasma Level

The mean daily MPA dose taken throughout the study period was 951 ± 478 mg/day, with a mean MPA trough level of 2.64 ± 2.28 µg/mL. The MPA dosage taken orally was significantly associated with the MPA plasma level, as measured using the HPLC. With each increase in MPA dosage by one point [mg], the MPA plasma level increased by 0.0022 µg/mL (95% CI 0.0018 to 0.0026), *p* = 0.001.

#### 3.3.2. MPA Plasma Level and Graft Rejection

There was no significant association between MPA plasma level and kidney graft rejection (Table 2). With each increase in the MPA plasma level by one point [µg/mL], the odds ratio for graft rejection decreased by 0.9957 (95% CI 0.8798 to 1.1268; *p* = 0.945). A similar result was obtained in the multivariable analysis adjusted for sex, age, donor-specific antibodies, and number of kidney transplantations: 0.9750 (95% CI 0.86222 to 1.10247; *p* = 0.686) (Table 3).

### 3.4. Pharmacodynamic Testing

#### 3.4.1. Predose IMPDH Enzyme Activity in PBMCs and Graft Rejection

In the pharmacodynamic analysis, there was a highly significant association between the predose IMPDH enzyme activity in PBMCs and kidney graft rejection (Table 2). 

With each increase in the predose IMPDH enzyme activity by one point [pmol/h], as measured at the time of graft rejection, the odds ratio for graft rejection increased by 1.013 (95% CI 1.007 to 1.019; *p* = 0.001). In the multivariable analysis adjusted for sex, age, donor-specific antibodies, and number of kidney transplantations, the corresponding result was 1.015 (95% CI 1.008 to 1.023; *p* = 0.001) (Table 3). 

Furthermore, a receiver operating characteristic (ROC) analysis was conducted to assess whether the predose IMPDH activity was a valuable marker for the diagnosis of BPR. The optimal predose IMPDH cut-off to distinguish between BPR and non-rejection was 93.3 [95% CI 82.7 to 103.8] pmol/h, with a sensitivity and a specificity of the assay of 0.58 and 0.68 and an AUC of 0.63 [95% CI 0.57 to 0.69; *p* < 0.01]. 

#### 3.4.2. Baseline (Pretransplant) IMPDH Enzyme Activity in PBMCs and Graft Rejection

There was no significant association between the baseline IMPDH activity (IMPDH enzyme activity before transplantation) in PBMCs and renal graft rejection (Table 2). With each increase in the baseline IMPDH enzyme activity by one point [pmol/h], the odds ratio for graft rejection increased by 1.003 (0.991 to 1.015); this was not significant either in the univariate (*p* = 0.649) or in the multivariable analysis (*p* = 0.610) (Table 3).

#### 3.4.3. IMPDH Increase (% of IMPDH Baseline) and Graft Rejection

An IMPDH increase (% of IMPDH baseline) was defined as a change in IMPDH enzyme activity at the time of biopsy/rejection in relation to the baseline IMPDH activity before transplantation.

A significant association between IMPDH increase (% of IMPDH baseline) and biopsy-proven rejection was detected. With each increment in IMPDH increase (% of IMPDH baseline) by one point, the odds ratio for graft rejection increased by 1.003 (1.001 to 1.006); this was significant in the univariate (*p* = 0.048) and multivariable analyses (*p* = 0.042) (Table 3).

## 4. Discussion

Currently, there is no established method to predict individual responsiveness to MPA therapy. In our study, we demonstrated that, with the presence of the IMPDH2 SNP rs11706052, the odds ratio for graft rejection increased by 1.81 in renal transplant patients. This was statistically not significant, but there was a significant result when a trend test was performed, indicating that there was a clear trend showing the association of SNP status with rejection risk. However, the sample size was too small to show statistical significance. In the future, pharmacogenetics may play an increasingly important role in individualized immunosuppressive therapy in the era of whole-genome sequencing, customizable peptide arrays, and genome-wide association studies [44].

In recent years, a number of studies on MPA pharmacokinetics in renal transplant patients have been published [45,46,47,48,49,50,51,52]. A therapeutic window of 1.0–3.5 µg/mL for MPA trough levels [33,53] and 30–60 mg/h/L for MPA AUC [39] may be targeted to maximize the therapeutic efficacy of MPA therapy and minimize the side effects [14]. However, since full AUC monitoring over 12 h is particularly labour-intensive and, therefore, impractical in routine clinical practice, limited sampling strategies or maximum a posteriori Bayesian estimation have been proposed [54,55,56,57,58,59,60]. In our study, for practical reasons, we focused on therapeutic drug monitoring using MPA trough levels. However, no significant association between the plasma MPA levels and graft rejection was found.

Pharmacodynamic monitoring to measure IMPDH enzyme activity is an advanced approach to individualize MPA therapy, as it better mirrors the biological response to MPA therapy than single surrogate parameters of MPA pharmacokinetics [61]. On the other hand, measurement of IMPDH activity is even more labour-intensive. This is complicated by the fact that the activity of IMPDH, which is inhibited by MPA, is not stable, but fluctuates between patients and even within the same individual [34,61]. Over 8-fold differences in basal IMPDH activity have been described [19,62]. Increased IMPDH enzyme activity pre-transplant has been associated with acute kidney transplant rejection [19]. Furthermore, IMPDH activity fluctuations become more evident after the start of MPA therapy, and there are inconsistent data on IMPDH activity during the early period of transplantation in the literature [6,63,64].

To measure IMPDH enzyme activity, peripheral blood mononuclear cells are the matrix of choice. Several studies including patients under MPA therapy have reported that MPA significantly decreases IMPDH levels in isolated PBMCs, while an unexpected rise in IMPDH activity in whole blood has been described [22]. For the determination of IMPDH, high-performance liquid chromatography (HPLC) is the method of choice in therapeutic drug monitoring [22,53].

Although the limitations of pharmacodynamic monitoring of IMPDH by means of HPLC are apparent (high methodological workload; the need for the isolation of PBMCs as the matrix of choice; low levels of activated lymphocytes in isolated PBMCs; no distinction between the enzyme activity of IMPDH1 and that of IMPDH2; and significant individual variations in IMPDH activity over time) [6], it could be shown in our study that the renal transplant patients with biopsy-proven graft rejection under MPA therapy have significantly higher predose IMPDH activity levels compared to the controls without rejection. It could be further shown that the patients with increased IMPDH activity over time at the time of biopsy/rejection relative to the baseline IMPDH activity before transplantation were those patients with allograft rejection. These are important findings that underline the importance of pharmacodynamic monitoring of MPA therapy.

The main limitations of our study are the observational design and the time of MPA and predose IMPDH measurements (since MPA was given twice a day, MPA and IMPDH measurements were obtained 12 h after the last MPA intake). It has been shown in the literature that the maximum inhibition of IMPDH takes place briefly after MPA uptake, which coincides with the MPA plasma peak, and is succeeded by a return of IMPDH activity to predose levels within 3.5 to 11 h [61,65]. Therefore, the assessments of MPA levels and IMPDH activity at several time points post-dose may have been of additional interest to better monitor further variables, such as the enterohepatic recirculation, and to further specify the predictive significance of these values. However, our personnel resources were limited. 

On the other hand, there are also significant strengths of our study. We had a large study population of 277 kidney transplant patients with prospective follow-up, including blood analyses at multiple time points. In addition, our combined pharmacogenetic, pharmacokinetic, and pharmacodynamic analyses provide comprehensive insight into the potential individualization of MPA therapy.

## 5. Conclusions

In this prospective cohort study, we show that the renal transplant patients with biopsy-proven graft rejection have a significantly higher predose IMPDH activity compared to the controls without rejection. Furthermore, it is demonstrated that patients with increased IMPDH activity over time at the time of the graft biopsy relative to the baseline IMPDH activity before transplantation have a significantly higher risk of allograft rejection.

Our study suggests that pharmacodynamic monitoring of IMPDH activity in PBMCs using a liquid chromatography–mass spectrometry may contribute to improving the clinical outcomes of MPA therapy by tailoring the treatment to the metabolic variability of individual patients.

## Figures and Tables

**Table 1 biomedicines-10-02882-t001:** Baseline characteristics of Study Patients.

	Total	Patients with BPR	Patients without BPR	*p*-Value
Number of subjects	277	84	193	
**Characteristics**				
Age (years)	54.6 ± 14.1	52.2 ± 14.4	55.6 ± 13.9	0.08
Gender—male/female (%)	66/34	61/39	70/30	0.57
First KTX/multiple KTX (%)	83/17	86/14	82/18	0.43
Mismatch	3.0 ± 1.4	3.3 ± 1.3	3.0 ± 1.4	0.51
PRA latest	2.9 ± 13.8	6.2 ± 21.2	2.9 ± 13.9	0.40
DSA negative/positive at baseline (%)	82/18	82/18	82/18	0.99
EC-MPS/MMF (%)	78/22	82/18	77/23	0.74

Abbreviations: DSA, donor-specific antibodies; EC-MPS, enteric-coated mycophenolate sodium; MMF, mycophenolate mofetil; KTX, kidney transplantation; PRA, panel reactive antibodies.

**Table 2 biomedicines-10-02882-t002:** Pharmacogenetic, Pharmacokinetic, and Pharmacodynamic Parameters in Kidney Transplant Recipients with and without Graft Rejection.

	Rejection Group (*n* = 84)	Control Group (*n* = 193)	*p*-Value
**Pharmacogenetic analysis:**			
**SNP status (** **rs11706052)**			0.076
Wild-type—number (%)	60 (71.4)	161 (83.5)	
Heterozygous SNP—number (%)	22 (26.2)	30 (15.5)	
Homozygous SNP—number (%)	2 (2.4)	2 (1.0)	
**Pharmacokinetic analysis:**			
MPA trough level [µg/mL]	2.57 ± 2.15	3.31 ± 2.67	0.95
**Pharmacodynamic analysis:**			
Predose IMPDH activity [pmol/]	110.1 ± 50.2	95.2 ± 45.5	0.001
Baseline (pretransplant) IMPDH activity [pmol/]	73.9 ± 16.8	71.5 ± 23.5	0.65

Abbreviations: IMPDH, inosine monophosphate dehydrogenase; MPA, mycophenolic acid; SNP, single nucleotide polymorphism.

**Table 3 biomedicines-10-02882-t003:** Association between pharmacokinetic and pharmacodynamic variables and kidney graft rejection.

	Regression Coefficient (95% CI) with Graft Rejection Univariate	*p*-Value	Regression Coefficient (95% CI) with Graft Rejection Adjusted ^a^	*p*-Value
MPA trough level [µg/mL]	0.9957 (0.8798 to 1.1268)	0.945	0.9750 (0.8622 to 1.1025)	0.686
Predose IMPDH enzyme activity at time of biopsy (pmol/h)	1.0130 (1.0070 to 1.0190)	0.001	1.0151 (1.0077 to 1.0225)	0.001
Baseline (pretransplant) IMPDH enzyme activity (pmol/h)	1.0028 (0.9906 to 1.0152)	0.649	1.0031 (0.9911 to 1.0153)	0.610
IMPDH increase (% of IMPDH baseline)	1.0028 (1.0001 to 1.0057)	0.048	1.0029 (1.0001 to 1.0057)	0.042

Abbreviations: IMPDH, inosine monophosphate dehydrogenase; MPA, mycophenolic acid. Multivariable regression analysis: ^a^ multivariable adjustment for sex (male/female), age (years), donor-specific antibodies, and number of kidney transplantations.

## Data Availability

The authors confirm that the data supporting the findings of this study are available within the article and/or its Appendix A.

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
