# Peer review of "Individualization of Mycophenolic Acid Therapy through Pharmacogenetic, Pharmacokinetic and Pharmacodynamic Testing"

_biomedicines, 2022, doi:10.3390/biomedicines10112882_

Round 1

Reviewer 1 Report

The work submitted for review is interesting and in line with current research trends. The work structure is correct. The results were properly discussed. Though the manuscript contains sufficient novelty to be accepted for publication, still minor modifications and suggestions are recommended to improve its quality.

1. Please precisely specify the transplant center, the data taken for the study came from.

2. Line 83. What drug was administered to the patients? Please give the name, dose, and producer of the drug.

3. Since when did the patients start immunosuppressive treatment?

4. I suggest lines 268-289 be moved to the introduction of the article.

5. There are literature references after the full stop sign in the sentences.  Please, move the literature reference prior to the full stop.

Author Response

Dear Reviewer, please find our point-by-point response to your comments in the Word file attached.

With best regards,
Wolfgang Winnicki

Reviewer 2 Report

biomedicines-2000940 entitled “Individualization of mycophenolic acid therapy through pharmacogenetic, pharmacokinetic and pharmacodynamic testing” studied the MPA from different clinical angles and found that IMPDH is a good biomarker indicating the potential risks of allograft rejection and can be used to monitor the therapeutic activity of MPA. The story would be interested to clinical research. But the following items must be either added or clarified:

1.      Section 2.2, please state the dose of MPA gave to the patients.

2.      Section 2.6, line 112, please correct the column information to 100 × 4.6 mm, instead of 100 – 4.6 mm. Line 113, please state the concentration of used phosphate buffer.

3.      Section 2.7.1, the authors stated “Calibrators, quality controls (QC) and the reaction mix (RM), consisting of …… were prepared, aliquoted and stored at -80°C, and for each assay fresh aliquots were used throughout this study”. Since all aliquots were prepared freshly, please explain the purpose of storing aliquots at -80°C.

4.      In section 2.7.1, please use a constant description of the units of molar concentration described, either use full unit name (e.g. mol/L) or abbreviation (nM).

5.      In section 2.7.1, Please explain how the XMP at the concentration of 7.5 nmol/μL was diluted to 375pmol with a factor of 1:3.413.

6.      In section 2.7.2, line 141, please use the correct symbol to emphasize t0.

7.      In section 2.7.2, lines 146-147, the authors state the supernatant was diluted and mixed with mobile phase A (2mM AmF, pH3.45). However, through the entire manuscript, there was no place where mentioned the function of this reagent.  Please rephase the sentence and provide the full name of AmF.

8.      In section 2.7.3, Please delete the symbols used to bracket “Luna Omega 1.6 μm Polar C18 100”. And please also correct the column information which the description in the manuscript was very confused. A typical description of a column is Luna Omega Polar C18 (100× 1.0 mm, 1.6 µm particles), for instance.

9.      In section 2.7.3, please provide a schematic sketch to show how the column was set within the LC-MS system while stayed in a cooler water bath. And please also explain how the column temperature stayed thermostatic at 18°C. Additionally, please check the spelling of the HPLC model name and provide the model’s name of mass spectrometry.

10.   Please rephase the LC-MS method in section 2.7.3 between lines 155 and 161. Make sure the compositions of mobile phases were introduced and consistently use either full name (i.e. mobile phase A), or abb (i.e. MPA) for all related mobile phases through the entire paragraph.

11.   Line 164, please use [M+H] to state the protonated molecular Ion.

12.   Please expand the description of the section 3.3.1 by introducing the doses of MPA which were tested and the respective MPA plasma levels.

Author Response

(The authors gave the same response as above.)
